# NEURAL TTS STYLIZATION WITH ADVERSARIAL AND COLLABORATIVE GAMES

**Shuang Ma**
State University of New York at Buffalo
Buffalo, NY
shuangma@buffalo.edu

**Daniel McDuff**
Microsoft Research
Redmond, WA
damcduff@microsoft.com

**Yale Song**
Microsoft Cloud & AI
Redmond, WA
yalesong@microsoft.com

## ABSTRACT

The modeling of style when synthesizing natural human speech from text has been the focus of significant attention. Some state-of-the-art approaches train an encoder-decoder network on paired text and audio samples $\langle x_{txt}, x_{aud} \rangle$ by encouraging its output to reconstruct $x_{aud}$. The synthesized audio waveform is expected to contain the verbal content of $x_{txt}$ and the auditory style of $x_{aud}$. Unfortunately, modeling style in TTS is somewhat under-determined and training models with a reconstruction loss alone is insufficient to disentangle content and style from other factors of variation. In this work, we introduce an end-to-end TTS model that offers enhanced content-style disentanglement ability and controllability. We achieve this by combining a pairwise training procedure, an adversarial game, and a collaborative game into one training scheme. The adversarial game concentrates the true data distribution, and the collaborative game minimizes the distance between real samples and generated samples in both the original space and the latent space. As a result, our model delivers a highly controllable generator with disentangled representation. Benefiting from the separate modeling of style and content, our model can generate human fidelity speech that satisfies the desired style conditions. Our model achieves start-of-the-art results across multiple tasks, including style transfer (content and style swapping), emotion modeling, and identity transfer (fitting a new speaker's voice).[1]

## 1 INTRODUCTION

In the past few years, we have seen exciting developments in Text-To-Speech (TTS) using deep neural networks that learn to synthesize human-like speech from text in an end-to-end fashion. Ideally, synthesized speech should convey the given text content in an appropriate *auditory style* which we refer to as *style modeling*. Modeling style is of particular importance for many practical applications such as intelligent conversational agents and assistants. Yet, this is an incredibly challenging task because the same text can map to different speaking styles, making the problem somewhat under-determined. To this end, the recently proposed Tacotron-based approaches (Wang et al., 2018; Skerry-Ryan et al., 2018a) use a piece of reference speech audio to specify the expected style. Given a pair of text and audio input, they assume two independent latent variables: $c$ that encodes content from text, and $s$ that encodes style from the reference audio, where $c$ and $s$ are produced by a text encoder and a style encoder, respectively. A new audio waveform can be consequently generated by a decoder conditioned on $c$ and $s$, i.e. $p(x|c, s)$. Thus, it is straightforward to train the model that minimizes the log-likelihood by a reconstruction loss. However, this method makes it challenging for $s$ to exclusively encode style because no constraints are placed on the disentanglement of style

---

[1]Project webpage: https://researchdemopage.wixsite.com/tts-gan

from content within the reference audio. It makes the model easy to simply memorize all the information (i.e. both style and content components) from the paired audio sample. In this case, the style embedding tends to be neglected by the decoder, and the style encoder cannot be optimized easily.

To help address some of the limitations of the prior work, we propose a model that provides enhanced *controllability* and *disentanglement ability*. Rather than only training on a single paired text-audio sample (the text and audio are aligned with each other), i.e. $(x_{txt}, x_{aud}) \rightarrow \tilde{x}$, we adopt a pairwise training procedure to enforce our model to correctly map input text to two different audio references $(x_{txt}, x_{aud}^+, x_{aud}^-)$, i.e. $(x_{txt}, x_{aud}^+) \rightarrow \tilde{x}^+$; $(x_{txt}, x_{aud}^-) \rightarrow \tilde{x}^-$, where $x_{aud}^+$ is paired with $x_{txt}$, and $x_{aud}^-$ is unpaired (randomly sampled). Training the model involves solving an adversarial game and a collaborative game. The adversarial game concentrates the true joint data distribution $p(x, c)$ by using a **conditional GAN loss**. The collaborative game is built to minimize the distance of generated samples from the real samples in both original space and latent space. Specifically, we introduce two additional losses, the **reconstruction loss** and the **style loss**. The style loss is produced by drawing inspiration from image style transfer (Gatys et al., 2016), which can be used to give explicit style constraints. During training, the the generator and discriminator combat each other to match a joint distribution. While at the same time, they also collaborate with each other in order to minimize the distance of the expected sample and the synthesized sample in both original space and hidden space. As a result, our model delivers a highly controllable generator and disentangled representation.

## 2 BACKGROUND

TTS can be formulated as a cross-domain mapping problem, i.e. given the source domain $Src$ (text) and target domain $Trg$ (audio), we want to learn a mapping $F : Src \rightarrow Trg$ such that the distribution of $F(Src)$ matches the distribution $Trg$. When modeling style in TTS, $F$ shall be conditioned on a style variable, which can be specified in many forms such as a reference audio waveform or a label. Given $(x_{txt}, x_{aud})$, the goal is then to synthesize a new audio waveform that contains the textual content specified by $x_{txt}$ and the auditory style specified by $x_{aud}$. Tacotron-based systems (Wang et al., 2018; Skerry-Ryan et al., 2018a) solve this with a reconstruction loss by training on paired data $x_{txt}$ and $x_{aud}$. Here, we describe their solution via a conditional probabilistic model admitting two independent sources of variation: a content variable $c_{1:T}$ with $T$ words specified by text $x_{txt}$, and a style variable $s$ given by the reference audio $x_{aud}$. Given $(x_{txt}, x_{aud})$, we can sample:

$$\text{content}: \ c_{1:T} \sim q_\varphi(c_{1:T}|x_{txt}), \ \ \text{style}: \ s \sim q_\phi(s|x_{aud}), \ \text{and output} \ \tilde{x} \sim p_\theta(x|c_{1:T}, s), \quad (1)$$

$p_\theta(x|c_{1:T}, s)$ is a likelihood function described by a decoder network, $Dec$. We define deterministic encoders $Enc_c$ that maps $x_{txt}$ to their corresponding content components, and $Enc_s$ that parameterizes the approximate posterior $q_\phi(s|x_{aud})$. It is natural to consider the conditional likelihood to be written as $x \sim p_\theta(x|Enc_c(x_{txt}), \ Enc_s(x_{aud}))$, and the training objective could be maximizing the log-likelihood:

$$\mathbb{E}_{c_{1:T} \sim q_\varphi(c_{1:T}|x_{txt}), \ s \sim q_\phi(s|x_{aud})} \big[ \log p_\theta(x|c, s) \big] \quad (2)$$

In practice, the learned mapping $F$ should be *injective*, i.e. there should be a one-to-one correspondence between input conditions and the output audio waveform. However, we argue that training only on paired data with maximum likelihood objective is insufficient to learn this mapping. Unlike $x_{txt}$ that purely contains content components, $x_{aud}$ consists of both style components $s$ and other factors $z$, such as verbal content that matches with $x_{txt}$. Therefore, the model needs to be able to disentangle $s$ from $z$. Otherwise, in the case of training on paired data by maximum likelihood objective, the model could simply learn to copy the waveform information from $x_{aud}$ to the output and ignore $s$. When given the same $x_{txt}$ but different $x_{aud}$ to such a model, it may still map sample to the same $\tilde{x}$. In the following sections, we introduce a way to prevent this degenerate issue.

## 3 APPROACH

Our proposed approach combines adversarial and collaborative games to train a TTS stylization model. Our training procedure, illustrated in Figure 1 (a), can also be considered as the swapping of style components. After swapping, the content components of both observations will remain the same, while the sources of style will change, and be aligned with $x_{aud}^+$ and $x_{aud}^-$, respectively. We now explain the two games involved in training the proposed model.

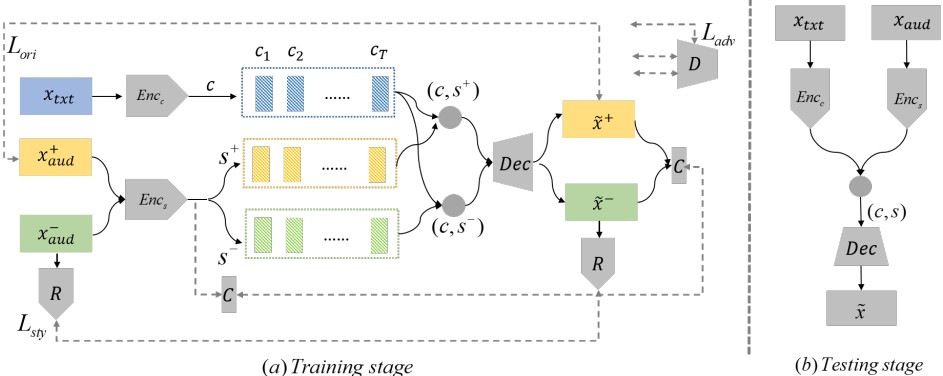

Figure 1: **Schematic diagrams of our model.** (a) A content encoder $Enc_c$ encodes $x_{txt}$ into a text embedding $c_{1:T}$, and a style encoder $Enc_s$ encodes both paired and unpaired audio samples, $x_{aud}^+$ and $x_{aud}^-$, into style embeddings $s^+$ and $s^-$, respectively. The decoder $Dec$ takes each of the two conditions $(c,\ s^+)$ and $(c,\ s^-)$, and generates $\tilde{x}^+ \sim p_\theta(x|c,\ s^+)$ and $\tilde{x}^- \sim p_\theta(x|c,\ s^-)$. All the losses involved in solving the adversarial game $(D)$ and collaborative game ($R$ and $C$) are indicated by dashed lines. (b) Given $x_{txt}$ and $x_{aud}$, we synthesize audio with $Enc_c$, $Enc_s$ and $Dec$. Note that $x_{aud}$ does not have to be paired with $x_{txt}$. See Appendix A for more details of each module.

### 3.1 ADVERSARIAL GAME

Because $\tilde{x}^-$ need not be aligned with the content factors of $x_{aud}^-$, we cannot enforce the reconstruction of $x_{aud}^-$. Instead, we enforce both $\tilde{x}^+$ and $\tilde{x}^-$ to be assigned high probabilities of belonging to the target domain using generative adversarial networks (GAN) Goodfellow et al. (2014). Specifically, we use a conditional GAN (Mirza & Osindero, 2014) to model a joint distribution of audio and content (i.e. $D(x,c)$), which provides a stronger constraint by enforcing the decision to be always conditioned on the content variable $c$. We define the min-max adversarial game:

$$\mathcal{L}_{adv} = \min_G \max_D \mathcal{L}_G + \mathcal{L}_D \tag{3}$$

$$\mathcal{L}_G = -\mathbb{E}_{c,s\sim s^+}\big[\log D(G(c,s),c)_3\big] - \mathbb{E}_{c,s\sim s^-}\big[\log D(G(c,s),c)_3\big] \tag{4}$$

$$\mathcal{L}_D = -\mathbb{E}_{c,s\sim s^+}\big[\log D(G(c,s),c)_1\big] - \mathbb{E}_{c,s\sim s^-}\big[\log D(G(c,s),c)_2\big] - \mathbb{E}_c\big[\log D(x_{aud}^+,c)_3\big] \tag{5}$$

Unlike the traditional binary classification setting (real or fake), we make $D$ a ternary classifier with $D(\cdot)_i$ representing the probability of $x$ being either "fake from paired input" ($D(\cdot)_1$), "fake from unpaired input" $D(\cdot)_2$, or "real audio sample" $D(\cdot)_3$. This ternary setting makes the discriminator more powerful because it must distinguish subtle differences between samples generated from paired and unpaired input. A similar setting has also been used in cross-domain image generation (Taigman et al., 2016). Our generator consists of two encoders $Enc_c$ and $Enc_s$ and a decoder $Dec$; the discriminator is used only during training.

### 3.2 COLLABORATIVE GAME

Although our adversarial game theoretically drives $p_\theta(x,c,s)$ toward the true data distribution, we find it to be insufficient to find the desired distribution, as there is little supervision from the observation what $s$ should represent. Especially for $x_{aud}^-$, the absence of a pairwise relationship makes it difficult to find the correct correspondence. As a result, $G$ might generate high-fidelity samples $\tilde{x}^-$ with incorrect $s$, and $D$ will still accept it as long as its style is different from $\tilde{x}^+$. Therefore, we impose explicit constraints on the generated samples with a style loss and a reconstruction loss.

**Style Loss.** In the computer vision literature, Gatys et al. (2016) captured the artistic style of an image using the gram matrix of features maps produced by a convolutional neural network. The gram matrix computes patch-level appearance statistics, such as texture, in a location-invariant manner. It is thus natural to expect that a gram matrix of feature maps computed from a mel-spectrogram

captures local statistics of an audio signal in the frequency-time domain, representing low-level characteristics of sound, e.g. loudness, stress, speed, pitch, etc. In fact, while the prosodic variation is often suprasegmental, certain characteristics, such as emotion, are captured by local statistics in the time-frequency domain. For example, Cheang & Pell (2008) have shown that a temporary reduction in the average fundamental frequency significantly correlates with sarcasm expression. More broadly, numerous past studies on prosody have been based on spectral characteristics, e.g. Dmitry Ulyanov (2016); Barry & Kim (2018).

Let $X$ and $\tilde{X}$ be the feature maps of the mel-spectrograms from the reference and the synthesized audio samples, respectively. We compute the gram matrices $W$ and $G$ as the inner-product of vectorized feature maps $X$ and $\tilde{X}$, respectively:

$$G_{i,j} = \sum_k \tilde{X}_{ik}\tilde{X}_{jk}, \quad and \quad W_{i,j} = \sum_k X_{ik}X_{jk} \tag{6}$$

Our style loss $L_{sty}$ is then the $L_2$ distance between $G$ and $W$ over all pairs of filters $i$ and $j$:

$$L_{sty}(G, W) = \frac{1}{N_f^2} \sum_{i,j} (G_{ij} - W_{ij})^2 \tag{7}$$

where $N_f$ is the number of filters. To produce the features maps, most existing approaches in image style transfer use the VGG-19 (Simonyan & Zisserman, 2014) pretrained on ImageNet (Russakovsky et al., 2015). However, mel-spectrograms look quite different from the natural images of the ImageNet, making the VGG-19 unsuitable for our work. We found that a simple four-layer CNN with random weights, denoted by $R$, perform just well for our purpose; similar findings have been reported recently by Ulyanov et al. (2017), showing that the structure of a CNN is sufficient to capture a great deal of low-level image statistics.

**Reconstruction Loss.** As $\tilde{x}^+$ is expected to be the same as $x_{aud}^+$, we include Eq. 2 and encourage the reconstruction in the original mel-spectrogram space:

$$\mathcal{L}_{ori} = -\mathbb{E}_{c\sim f(x_{txt}), s\sim g(x_{aud}^+)}(\log p_\theta(x|c, s^+)) \tag{8}$$

where $f(\cdot)$ and $g(\cdot)$ denote the deterministic encoding function of $Enc_c$ and $Enc_s$, respectively. We also encourage reconstruction in the latent space by introducing an inference network $C : x_{aud} \to z_c$ which approximates the posterior $p(z_c|x_{aud})$ as $z_c \sim p_c(z_c|x_{aud}) = C(x_{aud})$. $C$ reduces to an N-way classifier if $z_c$ is categorical. In our model, $C$ and $Enc_s$ share all layers and there is one final fully-connected layer to output parameters for the conditional distribution $p_c(z_c|x_{aud})$. To train $p_c(z_c|x_{aud})$, we define a collaborative game in the latent space:

$$\mathcal{L}_{lat} = \sum_{j=\{+,-\}} \left( -\mathbb{E}_{x\sim x_{aud}^j}[\log p_c(z_c|x)] - \mathbb{E}_{x\sim \tilde{x}^j}[\log p_c(z_c|x)] \right) \tag{9}$$

Minimizing the first term w.r.t. $C$ guides $C$ toward the true posterior $p(z_c|x_{aud})$, while minimizing the second term w.r.t. $G$ enhances $G$ with extra controllability, i.e. it minimizes the chance that $G$ could generate samples that would otherwise be falsely predicted by $C$. Note that we also minimize the second term w.r.t. $C$, which proves effective during training that uses synthetic samples to augment the predictive power of $C$. In summary, minimizing both $\mathcal{L}_{sty}$ and $\mathcal{L}_{rec}$ can be seen as a collaborative game between players $C$, $R$ and $G$ that drives $p_\theta(x|c, s)$ to match $p(x|c, s)$, and $p_c(z_c|x)$ to match the posterior $p(z_c|x)$, the reconstruction loss is thus given by:

$$\mathcal{L}_{rec} = \mathcal{L}_{ori} + \mathcal{L}_{lat} \tag{10}$$

### 3.3 IMPLEMENTATION DETAILS

We train our model with a combination of the GAN loss, style loss, and reconstruction loss:

$$L_{all} = L_{adv} + \alpha L_{sty} + \beta L_{rec} \tag{11}$$

We set $\alpha = 0.1$, $\beta = 10$ in our experiments. Our model is based on Tacotron (Wang et al., 2017b) that predicts mel-spectrograms directly from character sequences. The predicted mel-spectrogram can be synthesized directly to speech using either the WaveNet vocoder (van den Oord et al., 2016) or the Griffin-Lim method (Griffin & Lim, 1984). In our experiments, we use the Griffin-Lim for

fast waveform generation. For $Enc_c$, we use the same text encoder architecture of Skerry-Ryan et al. (2018b). The style encoder $Enc_s$ is a combination of *reference encoder* and *style token layers* proposed in Wang et al. (2018). We combine the encoded $s$ and $c$ as in Tacotron, i.e. for a content sequence $c$ of length $L$, we concatenate the style embedding with each state of the text embeddings. The inference network $C$ takes as input a style embedding and processes it through a fully-connected layers followed by batch normalization and Relu is added on top of each convolution layer. The output is mapped to an N-way classification layer. $R$ is a 2-D fully-convolution neural network with four layers, with filter dimensions of 32, 32, 64, 64, respectively. The discriminator $D$ has the similar architecture with the reference encoder, except that a style and content fusion unit is added before it, such that the predicted spectrogram and content information are jointly fed into the network. Finally, a fully-connected layer followed by ReLU maps the output to a 3-way classification layer. Note that, instead of using the character level content embedding $c_{1:T}$, here we adopt the *global sentence embedding*, which is the average of hidden unit activation over the sequence. A detailed diagram can be seen in Appendix 5. We train our model with a minibatch size of 32 using the Adam optimizer; we iterated 200K steps for EMT-4 and 280K steps for VCTK datasets. During training, $R$ is fixed weights. For testing, $C$, $R$ and $D$ are not needed, and we simply send a $\langle text, audio \rangle$ pairs into the model (unpaired audios are not needed in the testing stage), which is shown in Figure 1.

## 4 RELATED WORK

**Text-To-Speech (TTS)**: Recently, rapid progress has been achieved in TTS with end-to-end trained neural networks, e.g., WaveNet (van den Oord et al., 2016), DeepVoice (Arik et al., 2017), VoiceLoop (Taigman et al., 2018), Char2Wav (Jose Sotelo, 2017), (Saito et al., 2017; Saruwatari, 2018; Yang et al., 2017) and Tacotron (Skerry-Ryan et al., 2018b). Consequently, modeling *style* in TTS has become a subject of extensive study. DeepVoice2 (Gibiansky et al., 2017) and Deep-Voice3 (Ping et al., 2018) learn one or more lookup tables that store information about different speaker identities. However, they are limited to synthesizing voices of speaker identities seen during training. Unlike DeepVoice2 and DeepVoice3, Nachmani et al. (2018), which is based on VoiceLoop, can fit unseen speakers' voice at testing time. There is also a collection of approaches that are based on Tacotron, e.g., Tacotron-prosody (Wang et al., 2017a), prosody-Tacotron (Skerry-Ryan et al., 2018a) and GST (Wang et al., 2018). prosody-Tacotron uses an encoder to compute a style embedding from a reference audio waveform, where the embedding provides style information that is not provided by the text. The Global-Style-Token (GST) extends prosody-Tacotron by adding a new attention layer that captures a wide range of acoustic styles.

**Domain mapping by GANs**: Recently, GANs have shown promising results in various domain mapping problems. Cycle-GAN (Zhu et al., 2017a) and UNIT (Liu et al., 2017) perform image-to-image translation by adding a cycle-consistency loss to the learning objective of GANs. Further research has extended this to cross-domain translation. StackGAN (Zhang et al., 2016) generates images from text, and DA-GAN (Ma et al., 2018) operates across different domains, e.g., object transfiguration, human face to cartoon face, skeleton to natural object. Another line of work performs one-sided domain mapping without using the cycle consistency loss, e.g.,(Taigman et al., 2016). (Zhang et al., 2017) and (Hirokazu Kameoka, 2018) are mapping within text and speech domains. Moving beyond one-to-one domain mapping, Bicycle GAN Zhu et al. (2017b) maps samples from one domain to multiple target domains. Our work can also be considered as a one-sided cross-domain mapping that does not require cycle consistency, which makes the training more practical. We also follow the concept of Bicycle GAN that promotes a one-to-many mapping. To the best of our knowledge, ours is the first to formulate TTS as a cross domain mapping problem using GANs.

**Style transfer**: The recent success in image style transfer Gatys et al. (2016) has motivated approaches that model the acoustic style of sound using spectrogram. For example, Dmitry Ulyanov (2016) uses a simple 1-D convolutional layer with a ReLU to compute feature maps and then obtain style features by computing the gram matrix. Barry & Kim (2018) followed the same concept and adopted two different audio representations, the mel-spectrogram and the constant Q transform spectrogram. Inspired by this, in this work, we adopt the image style transfer concept to impose explicit style constraints on audio mel-spectrogram.

|  |  | EMT-4 | | VCTK | |
| --- | --- | --- | --- | --- | --- |
|  |  | Recon. error | WER | Recon. error | WER |
| Comparisons | prosody-Tacotron | 0.42 | 10.6 | 0.70 | 19.4 |
|  | GST | 0.61 | 10.2 | 0.77 | **18.1** |
|  | FaderNetwork | 0.59 | 11.1 | 0.78 | 19.0 |
|  | DeepVoice2 | – | – | 0.81 | 18.5 |
| Ablation study | $\mathcal{L}_{adv}$ | 0.73 | 11.1 | 0.81 | 19.2 |
|  | $\mathcal{L}_{adv}+\mathcal{L}_{sty}$ | 0.75 | 10.9 | **0.83** | 18.9 |
|  | GST+$\mathcal{L}_{sty}$+$\mathcal{L}_{rec}$ | 0.58 | 22.3 | 0.77 | 28.9 |
| Ours | $\mathcal{L}_{adv}+\mathcal{L}_{sty}+\mathcal{L}_{rec}$ | **0.76** | **10.2** | **0.83** | 18.2 |
| Groundtruth |  | – | 7.1 | – | 9.8 |

Table 1: Experimental results on disentanglement ability and controllability.

## 5 EXPERIMENTS

We evaluate our model from three perspectives: **content vs. style disentanglement ability** (Sec. 5.1), **effectiveness of style modeling** (Sec. 5.2), and **controllability** (Sec. 5.3). We use two datasets: **EMT-4**, an in-house dataset of 22,377 American English audio-text samples, with a total of 24 hours. All the audio samples are read by a single speaker, in four emotion categories: happy, sad, angry and neutral. For each text sample, there is only one audio sample labeled with one of the four emotion styles. **VCTK**, a publicly available, multi-speaker dataset containing recordings of clean speech from 109 speakers, with a total of 44 hours. As the raw audio clips have different specifications, we preprocess them by downsampling the audio to 24kHz and trimming leading and trailing silence, reducing the median duration from 3.3 seconds to 1.8 seconds.

We compare our method with three state-of-the-art approaches: **prosody-Tacotron** (Skerry-Ryan et al., 2018a) is similar to our model but trained on the reconstruction loss only. The style embeddings are obtained from the reference encoder directly. **GST** (Wang et al., 2018) incorporates the *Global Style Tokens* to prosody-Tacotron. **DeepVoice2** (Gibiansky et al., 2017) learns a look-up table capturing embeddings for different speaker identity. As DeepVoice2 is particularly designed for multi-speaker modeling, comparisons with DeepVoice2 is only performed on VCTK.

### 5.1 CONTENT VS. STYLE DISENTANGLEMENT ABILITY

**Reconstruction error of style embeddings**: If the style encoder $Enc_s$ has successfully disentangled style from other factors of variation in the audio input, we expect the style embedding $s$ to contain very little information about the content of the audio input. Therefore, we should expect poor performance when we try to reconstruct the audio sample purely from $s$. This motivates us to evaluate our model with the task of reconstructing audio samples from style embeddings. To this end, we train an autoencoder, where the encoder has the same architecture as $Enc_s$ and the decoder has six deconvolutional layers, with each layer having batch normalization and ReLU activation. To set the baseline, we first train the autoencoder from scratch using only the $L_2$ reconstruction loss; this results in the reconstruction error of 0.12. Next, we use precomputed style embeddings from different approaches and train only the decoder network using the reconstruction loss.

We report the results under the columns "Recon. error" in Table 5. prosody-Tacotron achieves the lowest reconstruction error, suggesting that the approach has the weakest ability to disentangle style from other factors in audio input. GST shows improvement over prosody-Tacotron, which demonstrates the effectiveness of the style token layer that acts as an information bottleneck from audio input to style embeddings. DeepVoice2 performs much better than both prosody-Tacotron and GST on the VCTK dataset. This shows the model particularly works well on modeling speaker identities. Compared to the three state-of-the-art approaches, our model performs the best on both datasets. We also evaluate the importance of different loss terms in our model. We can see that the adversarial loss $\mathcal{L}_{adv}$ provides a significant improvement over the baseline models, which suggests the effectiveness of our adversarial loss and pairwise training. When we add the style loss we get further improvements. We also remove the adversarial loss and add the style and reconstruction losses to the baseline GST; this produces even worse results. It is because, when training only on paired data with GST's encoder-decoder network, the reconstruction loss already imposes very strong supervision. In this case, additional constraints might on the contrary impaired the performance due to the risk of over fitting. While as our model is adversarially trained, the GAN loss regularizes the model, thus under this case, the style loss and reconstruction loss can help optimizing in a better way.

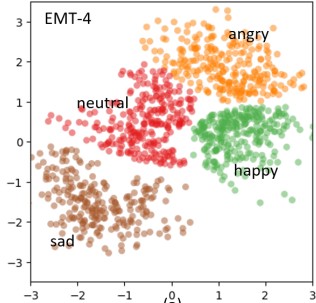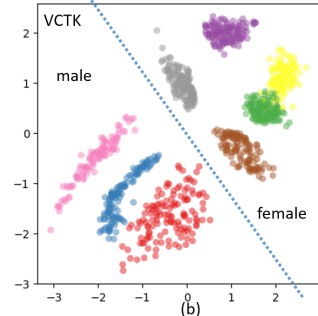

Figure 2: t-SNE visualization of the learned latent spaces for (a) EMT-4 and (b) VCTK datasets.

**Content and style swapping**: We conduct a qualitative evaluation where we generate audio samples by combining content and style embeddings from different ground-truth pairs of text and audio. Specifically, we randomly sample four (text, audio) pairs from the EMT-4 dataset, one for each emotion category, and create 16 permutations of text (content) and audio (style). Qualitative results are available on our project webpage. In in, the samples located on the diagonal are the results of **parallel style transfer**, i.e. reference audio sample is aligned with the text input. Off-diagonals are **unparalleled style transfer** where results in each column have the same content with different styles, and each row shares the same style with different content. By hearing our samples, we can see that the results are comparable for both parallel transfer and unparalleled transfer, which means the content and style components are disentangled. Even when transferring on two samples which are separated by a large distance in the latent space, e.g. sad to happy (row 2, line 4), the styles are correctly mapped (compare this to sad to sad (row 4, line 4)). We also conducted subjective study to ask seven subjects to classify these results by emotion category. The accuracy reaches 86%, which suggests the efficacy of our model in disentangling style and content components.

## 5.2 Effectiveness of Style Modeling

**Speaker style modeling**: We further evaluate the effectiveness of our approach on modeling styles by means of speaker verification. Specifically, we compare our style embeddings with the i-vector representation used in modern speaker verification systems (Kinnunen et al., 2017) on the speaker classification task. We report the results under the columns "Embeddings" in Table 5.3. We can see that, despite the fact that the i-vectors are specifically designed for speaker classification, our model can still achieve comparable results, which suggests that our model can produce generic feature representation for various auditory styles including speaker identity.

**Visualization of style embedding**: Figure 2 shows the t-SNE projection (van der Maaten & Hinton, 2008) of style embeddings from (a) the EMT-4 dataset and (b) the VCTK dataset. To create the plots, we randomly sampled 1,000 instances from each dataset: (a) 250 instances from each of the four emotion categories, and (b) 125 instances from 9 speakers (3 male and 5 female). We can see that the projections show clear boundaries between different style (emotion and speaker) categories. Interestingly, "sad" is far from the other three emotion categories; we believe that this is because sad speech usually have much lower pitch compared to other emotion categories. "neutral" is projected to the middle, which has roughly the same distance with other emotion categories. Also, we can see that there is a clear boundary between male samples and female samples.

## 5.3 Controllability

A good TTS system should allow users to control both content and style of the output. We consider two factors that affect the controllability: the **fidelity**, i.e. the synthesized speech should contain the desired content in a clearly audible form, and the **naturalness**, i.e. the synthesized speech should contain the desired style.

**WER of synthesized samples**: To validate the fidelity, we assess the performance of synthesized samples in a speech recognition task. We use a pre-trained ASR model based on WaveNet (van den Oord et al., 2016) to compute Word Error Rate (WER) for the samples synthesized by each model. Results are shown in Table 5. Our model performs comparably with, and sometimes even better than, the state-of-the-art approaches. Note that WER measures only the correctness of verbal content, not its auditory style. The results suggests that all the methods we have compared perform reasonably well in controlling the verbal content in TTS. When trained with more constraints on GST, i.e.

| | | Synthesized Audio | | | | Embeddings | |
|---|---|---|---|---|---|---|---|
| | | prosody-Tacotron | GST | DeepVoice2 | Ours | i-vector | Ours |
| EMT-4 | | 68% | 77% | – | 80% | – | – |
| VCTK | seen | 55% | 65% | 74% | 72% | – | – |
| | unseen | 51% | 62% | – | 70% | 75% | 71% |

Table 2: Classification accuracy for synthesized samples and learned style embeddings.

GST+$\mathcal{L}_{sty}$+ $\mathcal{L}_{rec}$, the performance gets worse. We suspect that this is because the autoencoder training procedure used in GST already gives strong supervision to the decoder. Thus, when added with more constraints, the model has overfitted to the training data. Our model does not have such problem because of the unpaired samples used during training, which act as strong regularizer.

**Classification accuracy on synthesized samples**: As the style we are modeling are all categorical, we evaluate the synthesized samples by a classification task. We train two classifiers on each dataset, which have 98% and 83% accuracy for EMT-4 and VCTK, respectively. We select 1000 samples synthesized from test set on EMT-4. To assess on VCTK, we test samples from both seen and unseen speakers, where 'seen speakers' mean the speakers are part of the training set, while the reference audio are selected from the test set. 'unseen speaker' means the speakers have never be seen during training, which means the model is asked to fit a new speaker's voice on testing stage. The results are shown in Table 5.3. As we can see, on the EMT-4 dataset, our model performs the better than prosody-Tacotron and GST. When tested on the seen data on VCTK, DeepVoice2 performs the best, but it fails to generalize to unseen speakers. Our model performs well in both cases.

**Style transfer**: To qualitatively evaluate Our model, we conduct style transfer. In this experiment, we want to compare our model against GST in how well they model varied styles in EMT-4. We randomly selected 15 sentences, where 10 of the sentences are from the test set, and 5 of them are picked on web (out of the dataset). To perform style transfer, we select four different reference audios from 'happy', 'angry', 'sad' and 'neutral', all of the reference audio samples are unseen during training. Each sentence is paired with these four reference audio samples for synthesizing, which will produce 60 new audio samples in total. The results can be found in our project page. We also compare our model against GST on the task of unparalleled transfer at scale. Specifically, we follow the same setting in Wang et al. (2018) to run side-by-side subjective study on 7-point ratings (-3 to 3) from "model A is the closest to the reference style" to "model B is the closest to the reference style", where model B is ours.

We recruited seven participants. Each listened to all 60 permutation of content and rated each set of audio style (emotions) comparing the result of our model versus the prosody-Tacotron model. They rated each pair of audio outputs on the 7-point scale. We performed a single-sample T-Test on the resulting ratings averaged across all participants. $\mu > 0$ means our model was judged as closer to the reference. Overall the emotion samples the participants rated our model as significantly closer to the reference ($\mu$=0.872, p$\ll$0.001). For each of the styles individually our model was consistently rated as significantly closer to the reference (neutral: $\mu$=0.295, p=0.01, happy: $\mu$=0.905, p$\ll$0.001, sad: $\mu$=1.646, p$\ll$0.001, angry: $\mu$=0.641, p$\ll$0.001). These results provide further evidence that our model can synthesize speech with the correct content and distinct auditory styles that are closer to the reference than the state-of-the-art comparison. We also evaluate the output using mean opinion score (MOS) naturalness tests. Our model reaches 4.3 MOS, outperforming 4.0 MOS reported in (Wang et al., 2018) and 3.82 MOS reported in (Skerry-Ryan et al., 2018a).

## 6 CONCLUSION

We propose an end-to-end conditional generative model for TTS style modeling. The proposed model is built upon Tacotron, with an enhanced content-style disentanglement ability and controllability. The proposed pairwise training approach that involves a adversarial game and a collaborative game together, result in a highly controllable generator with disentangled representations. Benefiting from the separate modeling of content $c$ and style $s$, our model can synthesize high fidelity speech signals with the correct content and realistic style, resulting in natural human-like speech. We demonstrated our approach on two TTS datasets with different auditory styles (emotion and speaker identity), and show that our approach establishes state-of-the-art quantitative and qualitative performance on a variety of tasks. For future research, an important direction can be training on unpaired data under an unsupervised setting. In this way, the requirements for a lot of work on aligning text and audios can be much released.

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

APPENDIX

## A    NETWORK ARCHITECTURES

We provide architecture details of our model. Figure 3 shows both the content and style encoder networks ($Enc_c$ and $Enc_s$, respectively) as well as the inference network ($C$). Figure 4 shows the decoder network ($Dec$), and Figure 5 shows the discriminator network ($D$). We further provide network parameter settings in the captions of each figure.

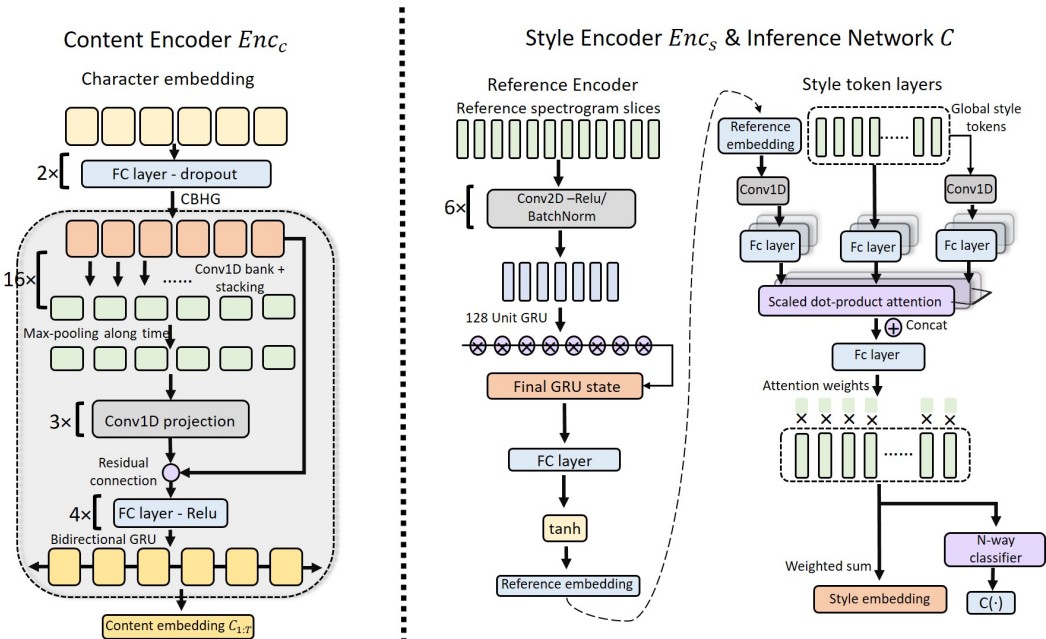

Figure 3: Encoder network architectures. **(Left) The content encoder ($Enc_s$):** A sequence of 128-D character-level embeddings is fed into two fully-connected layers which have [256, 128] units, respectively. Each layer is followed by a ReLU activation and dropout with a 50% chance. The output is fed into a CBHG block Wang et al. (2017b). Inside in the CBHG block, the Conv1D bank has 16 layers, where each layer has 128 units and comes with a ReLU activation. Next, the max-pooling layer has a stride of 1 and with a width of 2. The Conv1D projection has three layers, each with 128 units and a ReLU activation. After the residual connection is four fully-connected layers, each with 128 units and a ReLU activation. The final Bidirectional GRU has 128 cells. **(Right) The style encoder ($Enc_s$) and the inference Network ($C$):** The style encoder consists of a reference encoder and style token layers. The reference encoder takes a $N \times T_{mel} \times 80$ mel-spectrogram as input, where $N$ is batch size, $T_{mel}$ is length of mel-spectrogram, and 80 is the dimension. The six Conv2D layers have [32, 32, 64, 64, 128, 128] filters, respectively, each with a kernel size $3 \times 3$ and a stride of $2 \times 2$. Each layer is followed by a ReLU activation and batch normalization. Next is a single-layer GRU with 128 units. The final state from the GRU is fed into a fully-connected layer with 128 units and a tanh activation; this produces the reference embedding. In the style token layers, 10 global style tokens (GSTs) are randomly initialized. The reference embedding is used as a query for a multi-head attention unit. A learned linear weight is then output from the multi-head attention unit, and the style embedding is computed as a weighted sum. **The inference network ($C$)** shares the same architecture and parameters with $Enc_s$, except that a new N-way classifier (which consists of a fully connected layer followed by Softmas) is added on top.

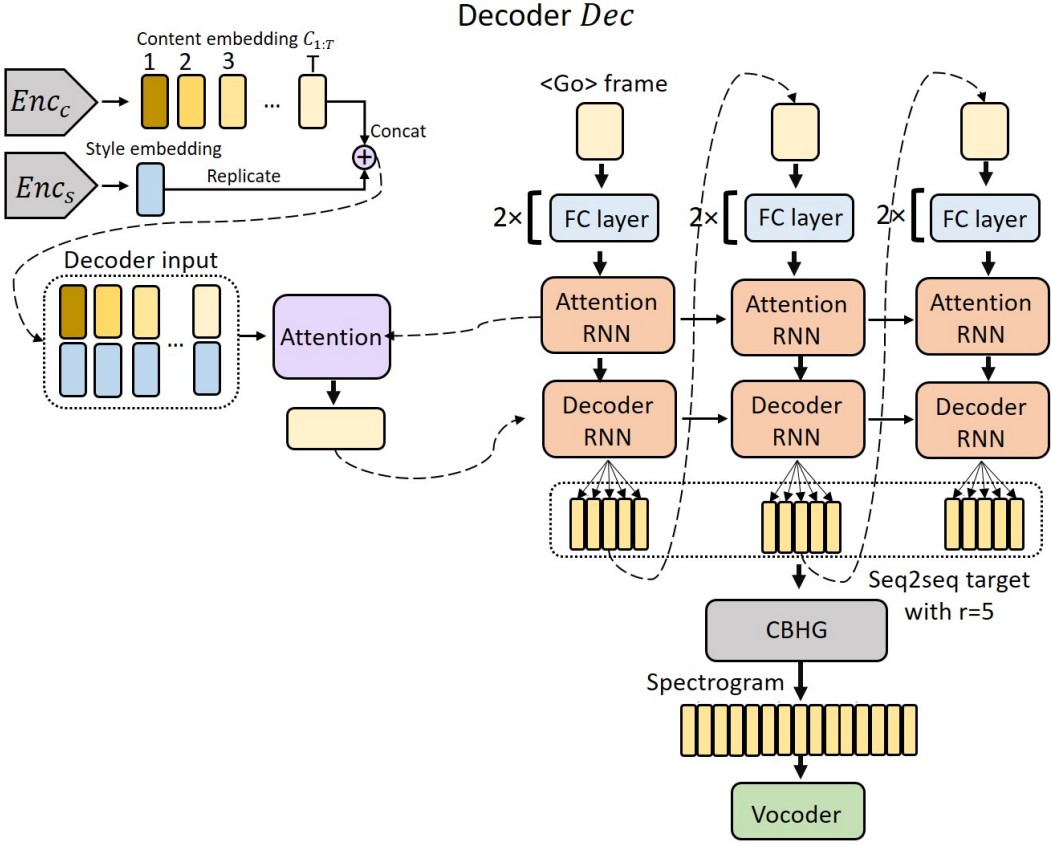

Figure 4: Decoder network architecture. The decoder ($Dec$) takes as input a concatenation of a content embedding sequence $C_{1:T}$ and the style embedding (replicated $T$ times). We then unroll each of the $T$ time slices by feeding them into two fully-connected layers, with [256, 128] units, respectively, followed by an attention RNN and a decoder RNN. The attention RNN has 2-layer residual-GRUs, each with 256 cells. The decoder RNN has a 256 cell one-layer GRU. As an output of each time step, 5 spectrogram slices are predicted ($r = 5$), and they are fed into the CBHG block (see Figure 3(left) for detail). The final output of the decoder is the predicted spectrogram. A vocoder is used to synthesize voice audios from the spectrograms. In this work, we use the Griffin-Lim algorithm Griffin & Lim (1984) to achieve fast waveform generation.

## Discriminator $D$

Figure 5: Discriminator network architecture. The main computation body is similar to the reference encoder (part of the style encoder in Figure 3), as shown on the right. The difference is that, instead of having only spectrograms as input, it has a combination of spectrograms (either ground truth or synthesized) and the content information (output from $Enc_c$), both shown on the left. Here we adopt global sentence embedding to represent the content information. The output content embedding from the *Content Encoder* $Enc_c$ is averaged along time over the whole sequence, which produces a $N \times 1 \times 128$ single embedding. To match with the dimension of the spectrogram, the single content embedding is replicated according to the spectrograms time step ($T_{mel}$), and they are concatenated together as the combined input.

# B ATTENTION PLOTS

In this section, we show attention plots of our model and a baseline model, comparing the robustness of these models for different lengths of the reference audio.

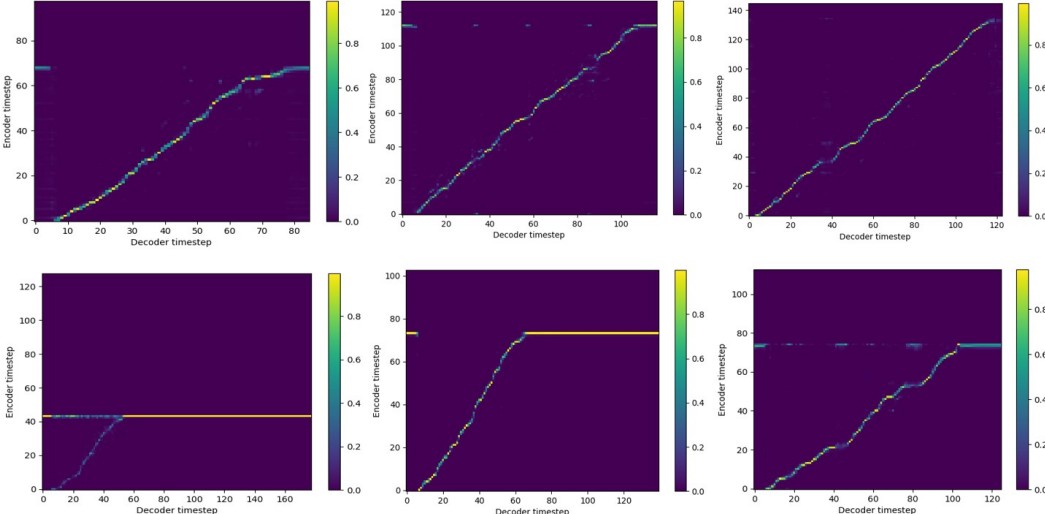

Figure 6: Attention alignments by different reference sentence lengths. From left to right, the sentence length are short, medium and long, respectively. The first row is obtained by using the *Style Encoder $Enc_s$* as shown in 3. The second row is obtained by only using the *Reference Encoder* (remove the *Style token layers in $Enc_s$*). As can be seen, adding the global style token layer made the network more robust to the variance in the length of the reference audio.

## C    MORE EXPERIMENT RESULTS

To better analyze the inference and attention mechanisms of our model, we further evaluate the performance in terms of the Word Error Rate (WER) and classification accuracy under different embedding sizes (Table 3) and different number of attention heads (Table 4).

Table 3: Ablation study on different reference embedding sizes.

| Embedding size | 32 | 64 | 128 | 256 | 512 |
|---|---|---|---|---|---|
| WER | 10.3 | 10.2 | 10.2 | 10.2 | 10.5 |
| Accuracy | 61% | 75% | 80% | 77% | 69 % |

Table 4: Ablation study on different number of heads in multihead attention.

| Attention heads | 2 | 4 | 8 | 16 |
|---|---|---|---|---|
| WER | 10.4 | 10.4 | 10.6 | 10.5 |
| Accuracy | 76% | 80% | 80% | 74% |

Table 3 shows the optimal embedding size is at 128; too small size (32) prevents essential information from flowing through the network, while too big size (512) leads to a poor ability to bottleneck the information from disentangling the style components with other factors within the reference audio. Also, the large embedding size means more parameters to optimize for, which results in the risk of over-fitting. Table 4 shows the results when applying difference numbers of attention heads. We get the best performance with four attention heads.

