# OpenReview forum: "Neural TTS Stylization with Adversarial and Collaborative Games"
_ICLR.cc/2019/Conference_

### Official Review · AnonReviewer1 · 2018-11-02
**Good adversarial domain adaptation ideas for TTS - but details on architecture needed**

**Rating:** 7
**Confidence:** 5

**Review:**

This paper proposes a method to synthesize speech from text input, with the style of an input voice provided with the text. Thus, we provide content - text - and style - voice. It leverages recent - phenomenal - progress in TTS with Deep Neural Networks as seen from exemplar works such as Tacotron (and derivatives), DeepVoice, which use seq2seq RNNs and Wavenet families of models. The work is extremely relevant in that audio data is hard to generate (expensive) and content-style modeling could be useful in a number of practical areas in synthetic voice generation. It is also quite applicable in the related problem of voice conversion. The work also uses some quite complex - (and very interesting!) - proposals to abstract style, and paste with content using generative modeling. I am VERY excited by this effort in that it puts together a number of sophisticated pieces together, in what I think is a very sensible way to implement a solution to this very difficult problem. However, I would like clarifications and explanations, especially in regards to the architecture.

Description of problem: The paper proposes a fairly elaborate setup to inject voice style (speech) into text. At train time it takes in text samples $x_{txt}$, paired voice samples (utterances that have $x_{txt}$ as content) $s+$ and unpaired voice samples $s-$, and produces two voice samples $x+$ (for paired  <txt, utterance>) and $x-$ (for unpaired txt/utterance). The idea is that at test time, we pass in a text sample $x_{txt}$ and an UNPAIRED voice sample $x_{aud}$ and the setup produces voice in the style of $x_{aud}$ but whose content is $x_{txt}$, in other words it generates synthetic speech saying $x_{txt}$. The paper goes on to show performance metrics based on an autoencoder loss, WER and t-SNE embeddings for various attributes.

Context:  The setup seems to be built upon the earlier work by Taigman et al (2016) which has the extremely interesting conception of using a {\it ternary} discriminator loss to carry out domain adaptation between images. This previous work was prior to the seminal CycleGAN work for image translation, which many speech works have since used. Interestingly, the Taigman work also hints at a 'common' latent representation a la UNIT using coupled VAE-GANs with cycle consistency (also extremely pertinent), but done differently. In addition to the GAN framework by Taigman et al, since this work is built upon Tacotron and the GST (Global Style Tokens) work that followed it, the generative setup is a sophisticated recurrent attention based seq2seq model.

Formulation:
A conditional formulation is used wherein the content c (encoding generated by text) is passed along with other inputs in the generator and discriminator. The formulation in Taigman assumes that there is an invariant representation in both (image) domains with shared features. To this, style embeddings (audio) gets added on and then gets passed into the generator to generate the speech. Both c and s seem to be encoder outputs in the formulation. The loss components of what they call ‘adversarial’, ‘collaborative’ and ‘style’ losses.

Adversarial losses
The ternary loss for D consists of

Discriminator output from ‘paired’ style embedding (i.e. text matching the content of paired audio sample)
Discriminator output from ‘unpaired’ style embedding (i.e text paired with random sample of some style)
Discriminator output from target ground truth style. The paper uses x_+, so I would think that it uses the paired sample (i.e. from the source) style.

Generator loss (also analogous to Taigman et al) consists of generations from paired and unpaired audio, possibly a loose analogue to source and target domains, although in this case we can’t as such think of ‘+’ as the source domain, since the input is text.

Collaborative losses
This has two components, one for style (Gatys et al 2016) and a reconstruction component. The reconstruction component again has two terms, one to reconstruct the paired audio output ‘x+=x_audio+’ - so that the input content is reproduced -  and the other to encourage reconstruction of the latent code.

Datasets and Results:
They use two datasets: one, an internal ‘EMT-4’ dataset with 20k+ English speakers, and the other, the VCTK corpus. Comparisons are made with a few good baselines in Tacotron2, GST and DeepVoice2.

One comparison technique to test disentanglement ability is to compare autoencoder reconstructions with the idea that a setup that has learnt to disentangle would produce higher reconstruction error because it has learnt to separate style and content.

t-SNE embeddings are presented to show visualizations of various emotion styles (neutral, angry, sad and happy), and separation of male and female voices. A WER metric is also presented so that generations are passed into a classifier (an ASR system trained on Wavenet). All the metrics above seem to compare excellently (better than?) with the others.

Questions and clarifications:

(Minor) There’s a typo in page 2, line 2. x_{aud}^+ should be x_{aud}^-.

Clarification on formulation: Making the analogy (is that even the right way of looking at this?) that the ‘source’ domain is ‘+’, and the target domain is ‘-’, in equation (5), the last term of the ternary discriminator has the source domain (x_{aud}^+) in it, while the Taigman et al paper uses the target term. Does this matter? I would think ‘no’, because we have a large number of terms here and each individual term in and of itself might not be relevant, nor is the current work a direct translation of the Taigman et al work. Nevertheless, I would like clarification, if possible, on the discrepancy and why we use the ‘+’ samples.

Clarification on reconstruction loss: I think the way it is presented, equation (8) is misleading. Apparently, we are sampling from the latent space of style and content embeddings for paired data. The notation seems to be quite consistent with that of the VAE, where we have a reconstruction and a recognition model, and in effect the equation (8) is sampling from the latent space in a stochastic way. However, as far as I can see, the latent space here produces deterministic embeddings, in that c = f(x_{txt}) and s = g(x_{aud}^+), with the distribution itself being a delta function. Also, the notation q used in this equation most definitely indicates a variational distribution, which I would think is misleading (unless I have misinterpreted what the style tokens mean). At any rate, it would help to show how the style token is computed and why it is not deterministic.

Clarification on latent reconstruction loss: In equation (9), how is the latent representation ‘l’ computed? While I can intuitively see that the latent space ‘l’ (or z, in more common notation) would be the ‘same’ between real audio samples and the ‘+’, ‘-’ fake samples, it seems to me that they would be related to s (as the paper says, ‘C’ and ‘Enc_s’ share all conv layers) and the text. But what, in physical terms is it producing? Is it like the shared latent space in the UNIT work, or the invariant representation in Taigman? This could be made clearer with an block diagram for the architecture.

(Major) Clarification on network architecture
The work references Tacotron’s GST work (Wang et al 2018) and the related Skerry-Ryan work as the stem architecture with separate networks for style embeddings and for content (text). While the architecture itself might be available in the stem work by Wang et al, I think we need some diagrams for the current work as well for a high level picture. Although it is mentioned in words in section 3.3, I do not get a clear idea of what the encoder/decoder architectures look like. I was also surprised in not seeing attention plots which are ubiquitous in this kind of work. Furthermore, in the notes to the ‘inference’ network ‘C’ it is stated that C and Enc_s share all conv layers. Again, a diagram might be helpful - this also applies for the discriminator.

Clarification on stability/mode collapse: Could the authors clarify how easily this setup trained in this adversarial setup?

Note on latent representation: To put the above points in perspective, a small note on what this architecture does in regards to the meaning of the latent codes would be useful. The Taigman et al 2016 paper talks about the f-constancy condition (and 'invariance'). Likewise, in the UNIT paper by Ming-Yu Liu - which is basically a set of coupled VAEs + cycle consistency losses, there is the notion of a shared latent space. A little discussion on these aspects would make the paper much more insightful to the domain adaptation practitioner.

Reference: This reference - Adversarial feature matching for text generation - (https://arxiv.org/abs/1706.03850) contains a reconstruction stream (as perhaps many other papers) and might be useful for instruction.

Other relevant works in speech and voice conversion: This work comes to mind, using the StarGAN setup, also containing a survey of relevant approach in voice conversion. Although the current work is for TTS, I think it would be useful to include speech papers carrying out domain adaptation for other tasks.

StarGAN-VC: Non-parallel many-to-many voice conversion with star generative adversarial networks.
https://arxiv.org/abs/1806.02169

I would rate this paper as being acceptable if the authors clarify my concerns, and in particular, about the architecture. It is also hard to hard to assess reproducibility in a complex architecture such as this.

---

> ### Author Response · Authors · 2018-11-09
> **Thanks for your thoughtful reviews and valuable comments.**
>
> 1. Typo in p2 l2.
> Thanks, we fixed it.
>
> 2. Clarification on formulation:
> Thank you for pointing out the discrepency. We provide detailed explanation below. In short, there is a subtle yet important distinction: We use '+' samples to regularize within-domain mapping (between (c, x_aud^+) and \tilde{x}^+), while Taigman et al., (2016) use '-' to promote cross-domain mapping (between (c, x_aud^-) and \tilde{x}^-)).
>
> Taigman's work use a pretrained function f(.) to extract latent embeddings from both the source and the target domains, i.e., z_s = f(s), z_t = f(t). They then use a decoder to map these to the target distribution, producing s2t and t2t. The s2t drives cross-domain mapping, while the t2t regularizes within-domain mapping. They use a single function f(.) to compute the embeddings from both the source (real human face) and the target (emoji human face) because the two domains share certain structures and properties, e.g., a face has two eyes with eyebrows on top. This makes t2t -- within-domain mapping -- relatively easy compared to ours (see below on why); so they include the target term in the loss (Eqn 3 in [Taigman et al., 2016]) to further promote cross-domain mapping.
>
> In our work, making the analogy, the source domain is '(content, style+)' and the target is '(content, style-)'. Both domains consist of two input modalities (text and sound) with very different characteristics. So we use two functions to represent each domain: Enc_c and Enc_s. Unfortunately, this makes it difficult to even ensure that within-domain mapping is successful. So, to strengthen within-domain mapping we modify the last term of the tenary discriminator to have x_aud^+ instead of the target x_aud^-.
>
> 3. Clarification on reconstruction loss:
> Yes, both the content c = f(x_txt) and the style s = g(x_aud^+) embeddings are deterministic. The only stochasticity comes from the data distribution. We revised the notation in the paper; please take a look.
>
> 4. Clarification on latent reconstruction loss:
> We have revised our paper with network architecture details, including a block diagram of the Inference Network 'C' that computes the latent representation 'l'; see Figure 3. The inference network is simply the style encoder (Enc_s) with a new classifier on top (one FC layer followed by softmax); all the weights are shared between C and Enc_s except for the new classifier layer.
>
> We agree that 'z' is a more commonly used notation to represent latent codes. We have changed the notation in the paper; thanks for the suggestion!
>
> 5. Clarification on network architecture
> We have revised our paper with block diagrams of our network architecture as well as parameter settings used in our implementation (Figure 3 to 5). We have also included an attention plot (Figure 6), showing the robustness of our approach to the length of the reference audio.
>
> 6. Clarification on stability/mode collapse:
> In TTS stylization, when mode collapse happens the synthesized voice samples will exhibit the same acoustic style although different reference audio samples are provided. While it is difficult to entirely prevent the mode collapse from ever happening (as is common in GAN training), we have a number of measurements (i.e., different loss terms in our adversarial & collaborative game) to alleviate the issue and to improve stability during training. Our qualitative results show more diverse synthesized samples than Tacotron-GST when different reference audio samples are given, suggesting our work clearly improves upon the state-of-the-art.  Our learning curve (https://researchdemopage.wixsite.com/tts-gan/image) also suggests that training with our loss formulation is relatively stable, i.e., the three loss values seem to converge to a stable regime.
>
> 7. Note on latent representation:
> Perhaps the most important message we want to deliver is: We are improving upon content vs. style disentanglement in acoustic signals by means of adversarial & collaborative learning. Extracting ``acoustic styles'' such as prosody has been an extremely difficult task. The state-of-the-art GST achieves this with an attention mechanism. But, as we argue in our paper, their loss construction makes it difficult to ``wipe out'' content information from acoustic signals; this is also shown in their qualitative results where prosody style transfer fails when the length of the reference audio clip is different from what is appropriate for the content to be synthesized. Our novel loss construction enables careful conditioning of our model so that the two latent representations, content 'c' and style 's' embeddings, become more precise than the previous method could obtain. In particular, our paired and unpaired input forumation, and the adversarial & collaborative game makes our model better condition the latent space so that the content information is effectively ignored in style embedding vectors.
>
> 8. Reference:
> We have incorporated those references in our revision.

---

> > ### Comment · AnonReviewer1 · 2018-11-26
> > **Rebuttal makes things much clearer.**
> >
> > Thank you for the clarifications. I feel that the material is now much more convincing after seeing the architectural presentation. It is illuminating to note that one can break up content and style to capture their essence as can be seen in figures 2, 3, 4 and 5 in the appendix. Fig 2 uses multiheaded attention to compute similarity between ref. embedding and randomly initialized tokens - this seems to be a new addition to the previous GST works (Skerry-Ryan et al 2018 and Wang et al 2018).
> >
> > Overall, This work exhibits a very high level of application - attention based seq2seq modeling with Tacotron setup, and manipulating content and style with instructive use of techniques from the formulation to the architectures used .
> >
> > I rule this as a clear accept.

---

### Official Review · AnonReviewer2 · 2018-11-04
**Review of “TTS-GAN: A GENERATIVE ADVERSARIAL NETWORK FOR STYLE MODELING IN A TEXT-TO-SPEECH SYSTEM”**

**Rating:** 6
**Confidence:** 3

**Review:**

This paper proposes to use a generative adversarial network to model speaking style in end-to-end TTS. The paper shows the effectiveness of the proposed method compared with Takotron2 and other variants of end-to-end TTS with intensive experimental verifications. The proposed method of using adversarial and collaborative games is also quite unique. The experimental part of the paper is well written, but the formulation part is difficult to follow. Also, the method seems to be very complicated, and I’m concerning about the reproducibility of the method only with the description in Section 3.

Comments
- Page 2, line 2: x _{aud} ^{+} -> x _{aud} ^{-} (?)
- Section 2: $T$ is used for audio and the number of words.

---

> ### Author Response · Authors · 2018-11-09
> **Thank you for the comments.**
>
> Thank you for the comments. We have fixed the typos in our revision.

---

### Official Review · AnonReviewer4 · 2018-11-08
**Good technical ideas, but suffers from clarity issues and weak evaluation.**

**Rating:** 6
**Confidence:** 5

**Review:**

Overview: This paper describes an approach to style transfer in end-to-end speech synthesis by extending the reconstruction loss function and augmenting with an adversarial component and style based loss component.

Summary: This paper describes an interesting technical approach and the results show incremental improvement to matching a reference style in end-to-end speech synthesis.  The three-component adversarial loss is novel to this task.  While it has technical merit, the presentation of this paper make it unready for publication.  The technical descriptions are difficult to follow in places, it makes some incorrect statements about speech and speech synthesis and its evaluation is lacking in a number of ways.   After a substantial revision and additional evaluation, this will be a very good paper.

The title of the paper and moniker of this approach as “TTS-GAN” seems to preclude the fact that in the last few years there have been a number of approaches to speech synthesis using GANs.  By using such a generic term, it implies that this is the “standard” way of using a GAN for TTS.  Clearly it is not. Moreover, other than the use of the term, the authors do not claim that it is.

While the related works regarding style modeling and transfer in end-to-end TTS models are well described, prior work on using GANs in TTS is not.  (This may or may not be related to the previous point.)  For example, but not limited to:
Yang Shan, Xie Lei, Chen Xiao, Lou Xiaoyan, Zhu Xuan, Huang Dongyan, and Li Haizhou, Statistical Parametric Speech Synthesis Using Generative Adversarial Networks Under a Multi-task Learning Framework, ASRU, 2017
Yuki Saito, Shinnosuke Takamichi, Hiroshi Saruwatari, Text-to-speech Synthesis using STFT Spectra Based on Low- /multi-resolution Generative Adversarial Networks, ICASSP 2018
Saito Yuki, Takamichi Shinnosuke, and Saruwatari Hiroshi, Training Algorithm to Deceive Anti-spoofing Verification for DNN-based Speech Synthesis, ICASSP, 2017.

Section 2 describes speech synthesis as a cross-domain mapping problem F : S -> T, where S is text and T is speech. (Why a text-to-speech mapping is formalized as S->T is an irrelevant mystery.)  This is a reasonable formulation, however, this is not a bijective mapping.  There are many valid realizations s \subset T of a text utterance t \in S.  The true mapping F is one-to-many.    Contrary to the statement in Section 2, there should not be a one-to-one correspondence between input conditions and the output audio waveform and this should not be assumed.  This formalism can be posed as a simplification of the speech synthesis mapping problem.  Overall Section 2 lays an incorrect and unnecessary formalism over the problem, and does very little in terms of “background” information regarding speech synthesis or GANs.  I would recommend distilling the latter half of the last paragraph.  This content is important -- the goal of this paper is to disentangle the style component (s) from the “everything else” component (z)  in x_{aud} by which the resultant model can be correctly conditioned on s and ignore z.

Section 3.2 Style Loss: The parallel between artistic style in vision and speaking style in speech is misplaced.  Artistic style can be captured by local information by representing color choices, brush technique, etc.  Speaking style and prosodic variation more broadly is suprasegmental.  That is it spans multiple speech segments (typically defined as phonetic units, phonemes, etc.).  It is specifically not captured in local variations in the time-frequency domain.  The local statistics of a mel-spectrogram are empoverished to capture the long term variation spanning multiple syllables, words, and phrases that contribute to “speaking style”.  (In addition to the poor motivation of using low-level filters to capture speaking style, the authors describe “prosody” as “representing the low-level characteristics of sound”. This is not correct.)  These filter activations are more likely to capture voice quality and speaker identity characteristics than prosody and speaking style.

Section 3.2: Reconstruction Loss: The training in this section is difficult to follow.  Presumably, l is the explicit style label from the data, the emotion label for EMT-4 and (maybe) speaker id for VCTK.  It is a rather confusing choice to refer to this as “latent” since this carries a number of implications from variational techniques and bayesian inference.  Similarly, It is not clear how these are trained. Specifically, both terms are minimized w.r.t. C but the second is minimized only w.r.t G.  I would recommend that this section be rewritten to describe both the loss functions, target variables, and the dependent variables that are optimized during training.

Section 3.3 How are the coefficients \alpha and \beta determined?

Section 3.3 “We train TTS-GAN for at least 200k steps.” Why be vague about the training?

Section 3.3. “During training R is fixed weights” Where do these weights come from? Is it an ImageNet classifier similar with a smaller network than VGG-19?

Section 5: The presentation of results into Table 1 and Table 2 is quite odd.  The text material references Table 1 in Section 5.1, then Table 2 in Section 5.2, then Table 1 in Section 5.3 and then Table 2 again in Section 5.3.  It would be preferable to include the tabular material which is being discussed in the same order as the text.

Section 5: Evaluation.  It is surprising that there is no MOS or naturalness evaluation of this work.  In general increased flexibility of a style-enabled system results in decreased naturalness.  While there are WER results to show that intelligibility (at least machine intelligibility) may not suffer, the lack of an MOS result to describe TTS quality is surprising.

Section 5: The captions of Tables 1 and 2 should provide appropriate context for the contained data.  There is not enough information to understand what is described here without reference to the associated text.

Section 5.1: The content and style swapping is not evaluated.  While samples are provided, it is not at all clear that the claims made by the authors are supported by the data.  A listening study where subjects are asked to identify the intended emotion of the utterance would be a convincing way to demonstrate the effectiveness of this technique.  As it stands, I would recommend removing the section titled “Content and style swapping” as it is unempirical.  If the authors are committed to it, it could be reasonably moved to the conclusions or discussion section as anecdotal evidence.

Section 5.3: Why use a pre-trained WaveNet based ASR model?  What is its performance on the ground truth audio?  This is a valuable baseline for the WER of the synthesized material.

Section 5.3 Style Transfer: Without support that the subject ratings in this test follow a normal distribution a t-test is not a valid test to use here.  A non-parametric test like a Mann-Whitney U test would be more appropriate.

Section 5.3 Style Transfer: “Each listened to all 15 permutations of content”.  From the previous paragraph there should be 60 permutations.

Section 5.3 Style Transfer: Was there any difference in the results from the 10 sentences from the test set, and the 5 drawn from the web?

Typos:
Section 1 Introduction: “x_{aud}^{+} is unpaired” -> “x_{aud}^{-} is unpaired”
Section 2: “Here, We” -> “Here, we”
Section 5.3 “Tachotron” -> “Tacotron”

---

> ### Author Response · Authors · 2018-11-26
> **clarifications**
>
> Title
> We appreciate this point, and removed the TTS-GAN moniker as it is quite generic.
>
> Bijective mapping
> We agree that regular speech synthesis is not a bijective mapping problem, because it may result in multiple meaningful results. We also mentioned this in our paper (Sec. 1 ln 6-7). However, we want to clarify our claim, by saying ‘bijective’, we refer to style modeling in TTS (a conditional generation), i.e. given textual string and a reference audio sample, the synthesized audio should one-to-one correspond to the given conditions (content from text and style from reference audio). If it is not a bijective mapping, e.g. one-to-many mapping, then one textual string could map to different styles, which neglects our style condition (reference audio). We have also elaborated on our claim, which can be seen in Sec. 2 (last paragraph).
>
> Style loss
> With all due respect, we disagree with the reviewer that prosody cannot be captured in local variations in the time-frequency domain. In fact, certain prosodic characteristics, such as emotion, are captured by local statistics in the time-frequency domain. For example, Cheang and Pell (2008) have shown that a temporary reduction in the average fundamental frequency significantly correlates with sarcasm expression.
> More broadly, numerous past studies on prosody have been based on spectral characteristics, e.g. Wang (2015),  Soleymani, et al. (2018), Barry (2018).
> That being said, we do agree with the reviewer that prosodic variation is often suprasegmental. Therefore, our approach to capturing speaking style can only model those prosodic variations that are characterized by local statistics. We have made this point clear in our paper in Section 3.2.
> Cheang, Henry S., and Marc D. Pell. "The sound of sarcasm." Speech communication 50.5 (2008): 366-381.
> Kun-Ching Wang. “Time-Frequency Feature Representation Using Multi-Resolution Texture Analysis and Acoustic Activity Detector for Real-Life Speech Emotion Recognition” (2015).
> Sobhan Soleymani, Ali Dabouei, et al. “Prosodic-Enhanced Siamese Convolutional Neural Networks for Cross-Device Text-Independent Speaker Verification” (2018).
> Shaun Barry, Youngmoo Kim. “Style Transfer for Musical Audio Using Multiple Time-Frequency Representations”. (2018)
>
>
> Reconstruction loss
> First, we have changed ‘I’ to ‘z_c’ to represent the latent code.
> If z_c is categorical, then C could be a N-way classifier. So you are right, z_c is the emotion label for EMT-4, and identities for VCTK.
> ‘Latent’ is commonly used in encoder-decoder networks and generative work, we do not feel it is a confusing word.
> The training details are present in the last paragraph of this section. In Eq9, the first term is minimized over C and the second term is minimized over both C and G. The hyperparameters were empirically determined.
> Different datasets need different numbers of training steps (for EMT-4 we trained for 200k steps, while for VCTK, we trained our model for 280k steps).
> The detailed description of the weights and network architecture of R can be found in our paper (last paragraph in ‘Style Loss’ section and line 4-5 in page 5).
>
> Presentation of the tables
> Due the page limitation, we prefer to present our paper in a more compact way. However, we could move elements to the appendices if necessary.

---

> > ### Author Response · Authors · 2018-11-26
> > **evaluations**
> >
> > MOS
> > We do not think MOS is a must have metric in our paper. Other relevant papers for stylization in TTS, e.g. prosody-Tacotron also do not include a MOS evaluation.
> > We have performed a number of evaluations quantitatively and qualitatively and believe that these extensive evaluations are sufficient to validate our work. The most important thing is, in our paper, how to disentangle style and content such that the encoder learns to produce effective style latent codes is the most important claim. So other than some evaluation metrics used in regular TTS, we also performed a set of experiments that do not typically appear in TTS work.
> >
> > Table captions
> > Thanks for your suggestion, we will revise the captions.
> >
> > Swapping
> > The most important claim in our paper is the ability to disentangle content and style. We believe this experiment actually is most important evaluation in validating our claim. Similar experiment are typically performed in computer vision papers, e.g. ‘Disentangling factors of variation in deep representations using adversarial training, NIPS 2016’ (Fig.3).
> >
> > ASR model
> > The ASR model is just a tool to evaluate different methods, here we just compare the relative performance.  But your suggestions are good, we will add the ground truth WER in our paper.
> >
> > Style transfer
> > To validate that the test follows a normal distribution would require a large amount of subjective studies.  We followed the precedent in the most recent works (GST, and prosody-Tacotron).
> >
> > Permutations
> > Yes, you are right. Thanks for your carefully reading the paper, we will change this in our paper.
> >
> > Results
> > The results turn out that they are almost equivalent.
> >
> > Typos
> > Thanks, we will modify the typos in our paper.

---

> > > ### Comment · AnonReviewer4 · 2018-11-27
> > > **evaluations response**
> > >
> > > MOS:
> > > The comment was about assessment of naturalness of the resultant speech, not prescribing an MOS test specifically.    In general prosodic modifications lead to decreased quality.  Assessing how large this degradation is valuable in assessing this work.  For what it's worth, both the GST and prosody-tacotron papers would significantly benefit from this kind of evaluation and I find it surprising that they were omitted.
> > >
> > > "So other than some evaluation metrics used in regular TTS, we also performed a set of experiments that do not typically appear in TTS work."
> > > The main evaluation metrics for "regular TTS" are subjective tests that look at naturalness and to a lesser extent (given the state of the art) intelligibility. The typical tests are MOS, MUSHRA, or ABX (AXB, AXY).  The intelligibility dimension has been assessed by the WER evaluation.  Naturalness (or quality) has not.  It might be reasonable to claim that for this work TTS quality (measured in terms of naturalness) is not important, and is therefore not evaluated.
> > >
> > > "The most important claim in our paper is the ability to disentangle content and style. We believe this [swapping] experiment actually is most important evaluation in validating our claim."
> > > I agree that this is the most important claim of the paper and the most important evaluation.  This is why it is so surprising that there is no evaluation here.  Rather 16 examples are offered.  It would be reasonable to ask a human rater to assess the emotional content of the utterance as either neutral, happy, sad, angry.  This seems to be the most direct assessment of the claim of the paper.  Does the newly synthesized utterance contain the desired emotional information?  The style transfer evaluation is much more effective at demonstrating this.  The examples without evaluation are unconvincing.
> > >
> > > Style transfer:
> > > "To validate that the test follows a normal distribution would require a large amount of subjective studies.  We followed the precedent in the most recent works (GST, and prosody-Tacotron)."
> > > 1) you do not need to use a t-test.  There are non-parametric tests available (specifically the Mann-Whitney U-test) that do not assume that the observations follow a normal distribution.  Most (all?) statistical packages support this test. 2) the test used in GST is not described.  In prosody-tacotron a 95% confidence interval is described, but not a t-test.  I hope that the confidence interval is generated non-parametrically in that work.  Using a mean and standard deviation derived from observations that are not normally distribtued would have generated a biased estimate of the confidence intervals. 3) even if these papers did use an unsupported statistical test, the t-test is still not valid without confirmation that the analyzed ordinal subject responses follow a normal distribution (in most cases they do not).

---

> > > > ### Author Response · Authors · 2018-11-27
> > > > **naturalness, swapping and style transfer**
> > > >
> > > > Naturalness:
> > > > Thanks for your constructive comments. It is correct that our evaluation is focused on the disentanglement of style and content, rather than directly assessing the naturalness of the TTS results, because disentangling content/style is the major focus of our work. In hindsight, however, we do agree with your point that measuring the naturalness could have provided additional insights into how our model performs compared to the baseline TTS systems. We promise to add a MOS evaluation results in the final version of our paper.
> > > >
> > > >
> > > > Swapping:
> > > > We also agree with the reviewer on this point. We will add human classification results on the style swapping experiment.
> > > >
> > > >
> > > > Style transfer:
> > > > We appreciate your clarification on evaluation metrics for our subjective study. Yes, we do agree with your comments, and will modify our metric based on a non-parametric test.

---

> > > > > ### Comment · AnonReviewer4 · 2018-11-27
> > > > > **significant progress**
> > > > >
> > > > > I want to thank you, the authors for the significant amount of work that was made to the paper during this comment period.
> > > > >
> > > > > I've revised my assessment upwards on the basis of this effort.

---

> > ### Comment · AnonReviewer4 · 2018-11-27
> > **clarifications reponse**
> >
> > Bijective mapping:
> > Even conditioned on style tokens the mapping is not bijective.  A given text, with the same condition (style), can be produced as "angry" with different acoustic realizations.    The content, speaker and condition can all be transmitted and there are still valid variations of the realization.    This could be theoretically true, if the condition is considered to be a specific prosodic realization P of speaker A speaking utterance X, and the target of generating speaker B speaking utterance X with realization P.  However, 1) given the state of the art and understanding of prosody, it is very underdetermined, and not exactly useful.  it is underdetermined because we do not have a way of disentagling prosodic realization from speaker identity.  While we have some approaches to map from one speaker's pitch range to another, transformation of normal and affected speaking rhythm and voice qualities from one speaker to another are not well understood or all that thoroughly well studied.  And 2) it's also not clear that this is the desired mapping.  The goal is to retain the conditioning variable -- here a coarse description of affect.  The realization of speaker A speaking utterance X with "angry" prosody in and of itself is not unique.  Neither are the realizations of speaker B speaking utterance X with "angry" prosody.  Even if there is a theoretical bijective mapping based on a highly specified condition, the practical mapping that is being learned here is many-to-many.    The broader point is that the "Ideal" F is not even a function.  The target is a set, not a point, f(x_txt, x_aud) = {t \in trg_{txt, aud}} where trg_{txt, aud} is the set of all valid realizations of the text, txt, and conditioning information, aud, by the target speaker.
> >
> > Stepping back, the concern with maximum likelihood that is being raised is that the learned F may not be injective, i.e. that the learned function may map multiple elements of the domain to the same realization and completely ignore x_{aud}.  This is a fair concern.  One issue with the term bijective is that determines that F should also be surjective -- that every element in \hat{x} should be mappable from some x_txt and x_aud.  This aspect isn't addressed by the work.
> >
> > Making this discussion more constructive -- 1) consider removing the term "Ideally" from section 2.  The description here is much more practical than it is ideal.  2) consider replacing bijective with injective.  I believe it's more consistent with the problem that is being solved.
> >
> > Style loss:
> > My initial description of prosody was perhaps too pointed at addressing the (since deleted) statement in the previous draft that claimed that prosody was only the low-level characteristics.  Prosody does include local time-frequency elements -- particularly as they capture voice quality.  The previous point was that prosody (in its entirety) cannot be captured by these representation.  Prosody includes (but is not limited to) pitch (intonation), intensity, speaking rate/rhythm, and the use of pauses (usually, but not only to impact phrasing) as well as voice quality.  The use of pitch and intensity are primarily relevant in a suprasegmental context.  For example, in English(es), an absolute pitch observation carries very little information, but a rising or falling pitch contour (or contextualized within the speakers pitch range or register) can have significant information on the semantics pragmatics and paralinguistics of the utterance.  I did not mean to suggest that there isn't important information in the time-spectrum.  However, if you consider the literature on prosody as a whole you'll find that the relative value of local spectral content is much less relevant than suprasegmental content.  (This includes the references mentioned in the comment above. There are corresponding papers for each of the tasks (sarcasm recognition, emotion recognition, prosody in speaker recognition) that show that suprasegmental representations of prosody are more valuable that short time analyses.)

---

> > > ### Author Response · Authors · 2018-11-27
> > > **'bijective mapping' and 'style loss'**
> > >
> > > Bijective mapping:
> > > Thanks for the constructive suggestion. We do agree that the bijective constraint might be too strict; injective mapping could be more appropriate to illustrate our setting. We have incorporated your two suggestions into the new revision. (Note: Since this discussion was at the very last minute, which was past the rebuttal period, we could not upload the new version of the paper. But the change is already made and will be reflected in the final version.)
> > >
> > > Style loss:
> > > Thanks for the clarification. Yes, we do agree that prosody, in its entirety, cannot be captured using local statistics in the time-frequency domain. As we clarified above, our style loss is limited to capturing only certain elements of prosody. To reflect this, we have already removed our statement regarding style loss and prosody in the revision.

---

### Official Review · AnonReviewer3 · 2018-11-09
**lack of details and proper comparisons**

**Rating:** 6
**Confidence:** 5

**Review:**

This paper proposes to use GAN to disentangle style information from speech content. The presentation of the core idea is clear but IMO there are some key missing details and experiments.

* The paper mentions '....the model could simply learn to copy the waveform information from xaud to the output and ignore s....'
--  Did you verify this is indeed the case? 1) The style embedding in Skerry-Ryan et al.'18 serves as a single bottleneck layer, which could prevent information leaking. What dimension did you use, and did you try to use smaller size? 2) The GST layer in Wang et al.'18 is an even more aggressive bottleneck layer, which could (almost) eliminate style info entangled with content info.

* The sampling process to get x_{aud}^{-} needs more careful justifications/ablations.
-- Is random sampling enough? What if the model samples a x_{aud}^{-} that has the same speaking style as x_{aud}^{+}? (which could be a common case).

* Did you consider the idea in Fader Netowrks (Lample et al.'17)', which corresponds to adding a simple adversarial loss on the style embedding? It occurs to be a much simpler alternative to the proposed method.

* Table 1. "Tacotron2" is often referred to Shen et al.'18, not Skerry-Ryan et al.'18. Consider using something like "Prosody-Tacotron"?

* The paramerters used for comparisons with other models are not clear. Some of them are important detail (see the first point above)

* The author mentioned the distance between different clusters in the t-SNE plot. Note that the distance in t-SNE visualizations typically doesn't indicate anything.

* 'TTS-GAN' is too general as the name for the proposed method.

---

> ### Author Response · Authors · 2018-11-26
> **Thanks for your thoughtful reviews and valuable comments**
>
> Sorry, it misleads readers by saying in this way, we will modify our description in the paper.
>
>
> 1. To clarify, we were trying to communicate that when training on purely paired data the network can easily to memorize all the information from the paired audio sample, i.e. both style and content components.
> For example, given (txt1, aud1), the network memorizes that as long as given a txt1, the result should be aud1. In this case, the style embedding tends to be neglected by the decoder, and the style encoder cannot be optimized easily. During test stage, when given (txt1, aud2), the network still produces an audio sample very similar to aud1, and the ‘style’ is not learned well. Our experiments on style transfer validate this claim. When comparing with GST, our synthesized audio is closer to the reference style.
>
> 2. Through empirical experiments we found that randomly sampling is enough for training.
>
> 3. Thanks for your suggestion. When we started this work, that idea was our first basic attempt. But it turns out, by simply adding an adversarial loss on the latent space did not produce good results. The most severe problem is it is not robust to various length reference audio samples. When the reference audio is longer than the input, the synthesized samples tend to have long duplicate tails, or sometime noises. It severely impairs the audio quality.
> We suspect that, to satisfy the correct classification, the style embedding is squeezed into the same scale, which is not robust to varied length sequential signals. The Fader Network was used for processing images which are a fixed dimension, this method does not seem to work well for audio. Therefore, in our current model, we promote the disentanglement by paire-wise training, which means we do not need to add an adversarial loss directly on the latent space, but on the generated samples. Our results show that this leads to more robust outcomes for sequential signals for different lengths. We will clarify this in the paper.
>
> 4. Thanks. It is a good suggestion to replace Tacotron2 with Prosody-Tacrotron. We will modify this in our paper.
>
> 5. The hyperparameters for our model can be seen in our implementation details and Appendix.
> The parameters used for other methods are the same with their original work.
>
> 6. As in this experiment, we want to evaluate how well our model can learn the latent space. In other words, are the style embeddings produced by our model effectively representing any desired style. By showing the t-SNE visualization, we can see that, the latent space learned by our model can be well separated into clusters according to the testing data distribution. The same experiment was also done in GST (Wang et al).
>
> 7. We appreciate that TTS-GAN is quite general.  We are happy to change the name of the paper.

---

> > ### Comment · AnonReviewer3 · 2018-11-28
> > **improved indeed**
> >
> > I agree with the other reviewers that the paper is significantly improved compared to the last version. I appreciate the author's efforts!
> >
> > Two of my main concerns remain, however:
> > * Some of the comparisons depend on the setting of the key hyper-params in the prosody-tacotron and GST models.  E.g. the authors mention that the proposed model synthesizes speech with a closer speaking style to reference than GST does. Did you observe the same trend when you use a bigger reference embedding size or more heads in the multiheaded attention in GST?  (It doesn't need to consistently beat GST, etc. but a depiction of the performance trend, or trade-off, would be helpful to better understand the work)
> >
> > * I am surprised by the author's results with Fader network style adversarial loss for style transfer, which also contradicts to my own experience. I don't think that idea is specific to image. As an obvious baseline, at least, I think the authors should put some relevant discussions around it in the paper.
> >
> > Again, thanks for the authors' thorough comments and the willingness to change the paper contents!

---

> > > ### Author Response · Authors · 2018-11-28
> > > **Thank you for the constructive suggestions!**
> > >
> > > We sincerely appreciate your constructive suggestions!
> > >
> > > re: Ablation study on the reference embedding size and the number of heads in multihead attention
> > > We feel that this is a good suggestion! We agree that conducting more comprehensive analyses with different degrees of "information bottleneck" -- i.e., the size of reference embedding and the number of heads in the multihead attention -- will provide interesting insights on how our model behaves. We promise to include this ablation study in the Appendix.
> > >
> > > re: Fader Networks
> > > It is interesting to hear that, unlike our case, adding an adversarial loss for style transfer helped achieve better results (we'd be curious to see the results; do you have a paper we can check? was your experience based on images only or also based on audio signals?) Regardless, we feel that this will be an easy-to-add baseline approach; we will include this in the final version.

---

> > > > ### Comment · AnonReviewer3 · 2018-11-28
> > > > **thanks**
> > > >
> > > > Thanks for willing to include the ablation studies & new discussion. I trust the authors and will raise my old rating by one level.

---

### Public Comment · ~tao_xia1 · 2019-02-25
**mark**

Good paper ,make a mark.

---

### Meta-Review · Area_Chair1 · 2018-12-14
**Good contribution with nice results and analysis**

**Confidence:** 5
**Recommendation:** Accept (Poster)

**Metareview:**

The paper proposes using GANs for disentangling style information from speech content, and thereby improve style transfer in TTS. The review and responses for this paper have been especially thorough! The authors significantly improved the paper during the review process, as pointed out by the reviewers. Inclusion of additional baselines, evaluations and ablation analysis helped improve the overall quality of the paper and helped alleviate concerns raised by the reviewers. Therefore, it is recommended that the paper be accepted for publication.